# Assessment of Heavy Metals Contamination and Antimicrobial Drugs Residue in Broiler Edible Tissues in Bangladesh

**DOI:** 10.3390/antibiotics12040662

**Published:** 2023-03-28

**Authors:** Shaikh Mohammad Bokhtiar, Mohammad Rafiqul Islam, Md. Jisan Ahmed, Abdur Rahman, Kazi Rafiq

**Affiliations:** 1Bangladesh Agricultural Research Council (BARC), Dhaka 1215, Bangladesh; 2Livestock Division, Bangladesh Agricultural Research Council (BARC), Dhaka 1215, Bangladesh; 3Department of Pathology, Faculty of Animal Science and Veterinary Medicine, Sher-e-Bangla Agricultural University, Dhaka 1207, Bangladesh; 4Department of Dairy Science, Faculty of Animal Science and Veterinary Medicine, Sher-e-Bangla Agricultural University, Dhaka 1207, Bangladesh; 5Department of Pharmacology, Bangladesh Agricultural University, Mymensingh 2202, Bangladesh

**Keywords:** heavy metals, antimicrobial drugs residue, broiler meat, Bangladesh

## Abstract

There are substantial public health consequences when hazardous heavy metal contaminants and antimicrobial drug residues are present in broiler edible tissues. This study aimed to assess the concentration of antimicrobial drugs and heavy metals residues in broiler meat, bones and edible composites (combinations of liver, kidney and gizzard). Samples were collected from different types of broiler farms, broiler wet meat markets and supermarkets, covering all five divisions of Bangladesh. The antimicrobial drug and heavy metal residues were analyzed by uHPLC and ICP-MS, respectively. In addition, a cross-sectional survey was conducted among broiler meat consumers in the study areas to evaluate their attitude towards the consumption of broiler meat. The survey clearly stated that broiler meat consumers in Bangladesh have a negative attitude toward the consumption of broiler meat, although all respondents reported to eat broiler meat regularly. The antibiotic with the highest prevalence of residues in broiler edible tissues was oxytetracycline, followed by doxycycline, sulphadiazine and chloramphenicol. On the other hand, all collected broiler edible tissues contained chromium and lead, followed by arsenic. The fact of the matter is that the antimicrobial drugs and heavy metals residues were found to be below the maximum residue limit (MRL), except for the lead content. In addition, the broiler meat samples from supermarkets had lower levels of antimicrobial drugs and heavy metals residue compared to the broiler meat collected from various types of farms and broiler wet meat markets. Irrespective of the source, broiler meat was found to contain antimicrobial drugs and heavy metals residues below the MRL, except for lead, suggesting that broiler meat is safe for human consumption. Therefore, raising public awareness regarding misconceptions about broiler meat consumption among consumers would be warranted.

## 1. Introduction

Since it began in the 1990s, broiler chicken production in Bangladesh has grown steadily to become a large industry. In Bangladesh, broilers or meat-producing chickens represent 40% of all poultry production, with an average daily production of one crore one-day-old broiler chicks [1]. Broiler chicken meat is very popular among the general people in Bangladesh because it is affordable and widely available. Broiler meat is the cheapest protein source in Bangladesh and contains essential amino acids, vitamins and minerals that promote a healthy body growth and boost the body’s immune system. In Bangladesh, increased broiler farming is happening for a variety of reasons, including chicken’s rapid growth rate, ability to produce meat of the highest quality, a quick return of investment, low operating cost and minimal area requirements. This industry could have a significant impact on the GDP of the nation as well as on the empowerment of women and the rural economy [2].

There is growing concern about heavy metals as major contaminants in poultry meat, feed and other poultry products [3]. Due to food safety issues and the risk to human health through the food chain, exposure to heavy metals in food and water is a significant concern on a national and international level. Heavy metals bioaccumulated in meat or other sources of food eventually find their way into human body tissues [4]. Arsenic, cadmium, chromium, lead and mercury are among the main metals of public health concern due to their high toxic potency [5]. Empirical studies in Bangladesh have reported the presence of heavy metals in feeds [6,7,8,9], leading to bioaccumulation in animal bodies, an observation that has received a lot of attention from researchers [10].

Antibiotics were developed to prevent bacterial infections in humans, animals and plants [11]. The indiscriminate use of antibiotics for both preventive and therapeutic purposes has gained attention on a global scale due to their residual effects and subsequent risks to consumers’ health [2,12,13,14]. In addition, the increasing use of antibiotics as growth promoters poses increasing concerns for antibiotic resistance in both livestock and humans [15,16], and residual antibiotics are now present in meat, which also poses a risk to consumer health [17]. The presence of toxic heavy metals and antibiotics residues in the food chains may cause multiple organ damage even at low levels of exposure, carcinogenicity, mutagenicity, bone marrow toxicity, diarrhea, vomiting, coronary arterial blocking resulting heart attack, diabetes, intellectual impairment in young children as well as the development of antimicrobial-resistant pathogens [18]. During the recent ongoing COVID-19 outbreak, there has been disinformation regarding the assumption that broilers have the potential to spread the COVID-19 virus. As a result, the demand for wholesome broiler chicken meat has reduced among common people. On the other hand, some publication and social media platforms have spread misleading propaganda that broiler chicken meat contains hazardous contaminants including heavy metals, antibiotics and other harmful substances, which posed a serious public health hazard elsewhere [19]. Such types of misinformation have spread misconceptions about broiler meat among the consumers, who have reduced broiler meat consumption, leading to a reduction of broiler meat prices and resulting in a major detrimental effect on the broiler industry. With this fact in mind, the present research work was undertaken to evaluate the concentration of antibiotics and heavy metals in broiler chicken meat, bones, edible composites (combinations of liver, kidney and gizzard), to provide present scenario to the broiler meat consumers of Bangladesh.

## 2. Results

A total of 95 respondents (broiler meat consumers) participated in the survey. Among the respondents, 86.3% were male, and 13.7% were female (Figure 1). Twelve questions were asked to assess the attitude of broiler meat consumers towards broiler meat consumption. The survey revealed that 100% of the respondents consumed broiler meat, but their consumption patterns were different. About 96.8% of people liked to eat cooked broiler meat, 23% of people liked to eat grilled broiler meat, and 13.6% of people liked to eat fried broiler meat. All (110%) of the respondents reported thinking that eating broiler meat is beneficial because it contains proteins and vitamins. In contrast, 12.6% of the participants reported the belief that broiler meat has harmful effects on the consumer; about 68.4%, 89.5% and 22.1% of people appeared to think that antibiotics, hormones and harmful elements are used during broiler production, respectively. About 93.6% of the respondents said that eating broiler meat increases immunity, and 74.7% of them said that eating broiler meat can prevent COVID-19 (Figure 1). These data clearly stated that broiler meat consumers in Bangladesh have a negative attitude toward the consumption of broiler meat, although they usually eat broiler meat.

The results shown in Table 1 reveled that the antibiotic with highest prevalence of residue in broiler meat was oxytetracycline (74.49%), followed by doxycycline (73.47%), sulphadiazine (1.02%) and chloramphenicol (1.02%). in broiler bone, the antibiotic with highest prevalence of residue was doxycycline (57.14%), followed by oxytetracycline (37.76%) and sulphadiazine (18.37%). In edible composites, the antibiotic with highest prevalence of residue was oxytetracycline (73.74%), followed by doxycycline (72.73%) and sulphadiazine (1.01%). The concentrations of oxytetracycline, doxycycline, sulphadiazine, chloramphenicol in all edible tissues ranged from 0.00 to 780.34 µg/kg, 0.00 to 414.31 µg/kg, 0.00 to 14.00 µg/kg and 0.00 to 6.47 µg/kg, respectively. 

On the other hand, the prevalence of chromium and lead in broiler meat, bone and composite was found to be 100%. Meanwhile, the prevalence of arsenic in broiler meat, bone and composite was 91.84%, 97.96%, and 93.88%, respectively (Table 2). The concentrations of chromium, lead and arsenic in all collected broiler edible tissues ranged from 20.00 to 2700.00 µg/kg, 10.00 to 1510.00 µg/kg and 0.00 to 100.00 µg/kg, respectively.

### 2.1. Large Farms

The results of sample testing showed that 85.7% of the broiler meat samples from broiler farms in five districts of the country contained oxytetracycline, and 95% of them contained doxycycline. On the other hand, 100% of the samples showed the presence of arsenic, chromium and lead. The broiler meat was found to containe an average of 10.6 µg/kg of oxytetracycline, 11.3 µg/kg of doxycycline, 6.8 µg/kg of arsenic, 185.6 µg/kg of chromium and 263.8 µg/kg of lead (Figure 2A,B), values which were below the maximum residue limits [MRL: maximum residue level, 100 µg/kg for oxytetracycline, 100 µg/kg for doxycycline, 40 µg/kg for arsenic, 1000 µg/kg for chromium, 100 µg/kg for lead. Source: IAEA, FAO, EU standards], except for lead. On the other hand, the results of the bone samples analysis from broiler chickens in large farms showed the presence of oxytetracycline in 33.3% of the samples, doxycycline in 42.8% of the samples, sulfadiazine in 14.2% of the samples. In addition, 100% of the analyzed samples showed the presence of arsenic, chromium and lead. The broiler bones contained an average of 30.5 µg/kg of oxytetracycline, 21.1 µg/kg of doxycycline, 1.3 µg/kg of sulfadiazine, 8.5 µg/kg of arsenic, 412.3 µg/kg of chromium and 531.4 µg/kg of lead (Figure 3A,B), values which were below the maximum residue limits (MRL), except for the lead content. 

About 90.4% of broiler chicken composite samples were found to contain oxytetracycline, 90.4% doxycycline, 4.7% sulfadiazine, 95.2% arsenic, 100% both chromium and lead. The broiler composites contained an average of 21.2 µg/kg of oxytetracycline, 28.4 µg/kg of doxycycline, 0.5 µg/kg of sulfadiazine, 12.53 µg/kg of arsenic, 182.3 µg/kg of chromium and 281.9 µg/kg of lead (Figure 4A,B), values which were below the MRL, except for the lead content.

### 2.2. Medium Farms

Oxytetracycline was present in 76.1% of broiler chicken meat samples, doxycycline in 90.4% of the samples, sulfadiazine in 9.5% of the samples, and arsenic, chromium and lead in 100% of broiler chicken meat samples from broiler farms in five districts of the country. Broiler meat contained an average of 11.9 µg/kg of oxytetracycline, 10 µg/kg of doxycycline, 0.8 µg/kg of sulfadiazine, 7.3 µg/kg of arsenic, 151.9 µg/kg of chromium and 230.4 µg/kg of lead (Figure 2A,B), values which were below the maximum residue limits, except for lead. On the other hand, the broiler bone samples from medium farms showed the presence of oxytetracycline in 61.9% of the samples, doxycycline in 47.6% of the samples, sulfadiazine in 9.5% of the samples, arsenic in 95.2% of the samples and both chromium and lead in 100% of the samples. The broiler bones contained an average of 72 µg/kg of oxytetracycline, 19.6 µg/kg of doxycycline, 1.4 µg/kg of sulfadiazine, 9.2 µg/kg of arsenic, 540.9 µg/kg of chromium and 354.2 µg/kg of lead (Figure 3A,B), values which were below the maximum residue limits, except for the lead content. On the other hand, 85.7% of the broiler composite samples from medium farms contained oxytetracycline, and 76.1% doxycycline; in addition, 100% of the broiler composite samples contained arsenic, chromium and lead. The broiler composites contained an average of 15.3 µg/kg of oxytetracycline, 14.7 µg/kg of doxycycline, 10.9 µg/kg of arsenic, 186.6 µg/kg c of hromium and 271.9 µg/kg of lead (Figure 4A,B), values which were below the maximum residue limits, except for lead.

### 2.3. Small Farms

Oxytetracycline was present in 66.6% of the broiler chicken meat samples, doxycycline in 100% of the samples, and arsenic, chromium and lead in 100% of the samples. Broiler meat contained an average of 9.2 µg/kg of oxytetracycline, 11.2 µg/kg of doxycycline, 7.5 µg/kg of arsenic, 207.8 µg/kg of chromium and 236.1 µg/kg of lead (Figure 2A,B), values which were below the maximum residue limits, except for the lead content. On the other hand, the results of the broiler bone samples from small farms showed the presence of oxytetracycline and doxycycline in 42.8% of the samples, sulfadiazine in 33.3% of the samples, arsenic in 95.2% of the samples, and both chromium and lead in 100% of the samples. The broiler bones contained an average of 49.7 µg/kg of oxytetracycline, 31.2 µg/kg of doxycycline, 2.4 µg/kg of sulfadiazine, 9.8 µg/kg of arsenic, 505.2 µg/kg of chromium and 546.6 µg/kg of lead (Figure 3A,B), values which were also below the maximum residue limits, except for the lead content. About 71.4% of the broiler composite samples from medium farms had oxytetracycline, and 76.1% had doxycycline, while 100% of the broiler composite samples contained arsenic, chromium and lead. The broiler composite samples from small farms contained on average 18.6 µg/kg of oxytetracycline, 23 µg/kg of doxycycline, 14.5 µg/kg of arsenic, 203 µg/kg of chromium and 289 µg/kg of lead (Figure 4A,B), values which were below the maximum residue limits, except for the lead content.

### 2.4. Broiler Wet Meat Markets

Chloramphenicol was present in 3.4% of the samples of broiler chicken meat collected from broiler wet meat markets in five districts of the country, oxytetracycline in 58.6% of the samples, doxycycline in 89.6% of the samples, arsenic in 86.2% of the samples, and chromium and lead in 100% of the samples. Broiler meat contained an average of 0.22 µg/kg of chloramphenicol, 8.5 µg/kg of oxytetracycline, 11.9 µg/kg of doxycycline, 4.6 µg/kg of arsenic, 245.1 µg/kg of chromium and 360.6 µg/kg of lead (Figure 2A,B), values which were below the MRL. On the other hand, the results of the broiler bone samples analysis showed that 24.1% of the samples contained oxytetracycline, 79.3% of the samples contained doxycycline, 10.3% of the samples contained sulphadiazine, and 100% of the samples contained arsenic, chromium and lead. The broiler bones contained an average of 62.8 µg/kg of oxytetracycline, 35.9 µg/kg of doxycycline, 0.7 µg/kg of sulfadiazine, 4.4 µg/kg of arsenic, 556.5 µg/kg of chromium and 497.9 µg/kg of lead (Figure 3A,B), values which were below the MRL, except for the lead content. Oxytetracycline was present in 63.3% of the broiler composites samples obtained from broiler wet meat markets, doxycycline in 70% of the samples, arsenic in 90% of the samples, and both chromium and lead in 100% of the samples. The broiler composites contained, on average, 17.1 µg/kg of oxytetracycline, 16.3 µg/kg of doxycycline, 13.2 µg/kg of arsenic, 322.3 µg/kg of chromium and 366.9 µg/kg of lead (Figure 4A,B), values which were below the maximum residual limits, except for lead.

### 2.5. Supermarkets

We found that 16.6% of broiler meat samples collected from supermarkets in the Dhaka district contained oxytetracycline, 66.6% doxycycline, 50% arsenic, and 100% chromium and lead. Broiler meat contained an average of 1.2 µg/kg of doxycycline, 4.9 µg/kg of arsenic, 163.1 µg/kg of chromium and 205 µg/kg of lead (Figure 2A,B), values which were below the maximum residue limits. On the other hand, the broiler bone samples collected from supermarkets showed the presence of doxycycline in 100% of the samples, and arsenic, chromium and lead in 100% of the samples. The broiler bones contained an average of 27.5 µg/kg of doxycycline, 4.5 µg/kg of arsenic, 185 µg/kg of chromium and 393.3 µg/kg of lead (Figure 3A,B), values which were below the maximum residue limits, except for the lead content. Oxytetracycline and doxycycline were present in 16.6% of the samples of broiler composite obtained from supermarkets, arsenic in 50% of the samples, and chromium and lead in 100% of the samples. The broiler composites contained an average of 0.4 µg/kg of oxytetracycline, 3.8 µg/kg of doxycycline, 3.7 µg/kg of arsenic, 301.8 µg/kg of chromium and 328.3 µg/kg of lead (Figure 4A,B), values which were below the maximum residue limits, except for the lead content.

In addition, the concentration of chloramphenicol (Figure 5A), sulphadimidine (Figure 5B), oxytetracycline (Figure 5C), and doxycycline (Figure 5D) residue in broiler meat, bone and edible composite collected from various large farms, medium farms, small farms, broiler wet meat markets and supermarkets of the Dhaka district, Gazipur district, Barisal district, Chittagong district and Rajshahi district in Bangladesh are shown in Figure 5. Meanwhile, the concentration of arsenic, chromium, lead residue in broiler meat, bone, and edible composite are shown in Figure 6A,B,C, respectively.

### 2.6. Correlation

*Overall Correlation between the Levels of antimicrobials and Heavy metals*: there was a positive correlation between arsenic and sulphadiazine levels, which was significant, at the level of 0.05, and the correlation value was 0.114. In addition, the correlation between the levels of chromium and lead was 0.341 (*p* < 0.01), indicating a significant positive correlation between the levels of these contaminants. However, the correlation between the levels of arsenic and those of chromium and lead was −0.123 (*p* < 0.05) and −0.203 (*p* < 0.01), meaning that they were negatively correlated, which is significant (Appendix A).

*Correlation between the Levels of antimicrobials and Heavy metals found in Broiler meat:* there was a small positive correlation between the levels of arsenic and sulphadiazine, which was significant at the level of 0.05; the correlation value was 0.114. In addition, there was a medium correlation between the levels of chromium and lead, of 0.341 (*p* < 0.01), indicating a significant positive correlation between contaminants. However, the correlation of the levels of arsenic with those of chromium and lead was −0.123 (*p* < 0.05) and −0.203 (*p* < 0.01), indicating a small negative correlation, which is significant (Appendix A).

*Correlation between the Levels of antimicrobials and Heavy metals found in Broiler meat in Barisal:* there was a positive correlation (0.286) between oxytetracycline and lead in Barisal that was significant (*p* < 0.05). Moreover, the correlation between doxycycline and lead levels was also positive (0.348) and significant at the level of 0.01. However, there was a significant negative correlation between the levels of arsenic and those of doxycycline, arsenic and sulphadiazine corresponding to −0.320 and −0.308 (*p* < 0.05) (Appendix A).

*Correlation between the Levels of antimicrobials and Heavy metals found in Broiler meat in Dhaka:* there was a significant positive correlation between the levels of arsenic and those of oxytetracycline, which was 0.237, significant at the level of 0.05. In addition, the correlation between arsenic and sulphadiazine levels was also significant at the level of 0.05, corresponding to 0.275. The correlation of lead levels with chloramphenicol and doxycycline levels was 0.249 and 0.245, respectively, and was significant (*p* < 0.05). In addition, the correlation between the levels of chromium and those of lead was 0.462 at the level of 0.01 significance, thus stronger than the other correlation measured (Appendix A).

*Correlation between the Levels of antimicrobials and Heavy metals found in Broiler meat in Gazipur:* we found a moderate positive correlation between lead and chromium levels that was significant at the level of 0.01, which was 0.4. In addition, there was a correlation between sulphadiazine and arsenic levels, that was also a significant correlation (0.355) at the level of 0.05 (Appendix A).

*Correlation between the Levels of antimicrobials and Heavy metals found in Broiler meat in Rajshahi:* the correlation between lead and chromium levels in Rajshahi city was 0.365, i.e., a positive moderate correlation, which was also significant at the level of 0.01 (Appendix A).

*Correlation between the Levels of antimicrobials and Heavy metals found in Broiler meat in Large Farms:* A positive correlation is shown in the above-mentioned Table between lead and chromium levels. The correlation value was 0.524, indicating a moderate correlation at the significance level of 0.01. However, the correlation between arsenic and lead levels was negative. The value of the correlation was −0.356 (*p* < 0.01) (Appendix A).

*Correlation between the Levels of antimicrobials and Heavy metals found in Broiler meat in Medium Farms:* there was a negative correlation (−0.263) between arsenic and chromium levels, which was significant at the level of 0.05 (Appendix A).

*Correlation among antimicrobials and Heavy metals found in Broiler meat in Small Farm:* The correlation between Lead and Chromium is highly significant (*p* < 0.01) which is 0.416, a positive correlation between these two metals (Appendix A).

***Correlation between the Levels of antimicrobials and Heavy metals found in Broiler wet meat markets****:* The correlation between lead and chromium levels was highly significant (*p* < 0.01), corresponding to 0.312, indicating a positive correlation between the levels of these two metals (Appendix A). 

## 3. Methods and Materials

### 3.1. Study Area and Sample Collection

The samples were collected from farms, categorized into small (broiler chickens less than 2000), medium (broiler chickens from 2000 to less than 5000), and large (broiler chickens more than 5000) based on the number of broiler chickens in the shed, in the Dhaka, Chattogram, Gazipur, Rajshahi and Barisal districts, covering four divisions of Bangladesh. Samples were also collected from broiler wet meat markets in the Dhaka, Chattogram, Gazipur, Rajshahi and Barisal districts. In addition, samples were collected from supermarkets in Dhaka city. The area was selected considering the number of poultry farms, the number of hatcheries and the number of broiler consumers. A total of 295 samples of breast muscle (*n* = 98), bone marrow (*n* = 98) and composite (liver, kidney, gizzards) (*n* = 99) were collected from different types of farms, broiler wet meat markets from five districts of Bangladesh and supermarkets of Dhaka city. Out of these, 192 samples were collected from farms categorized as large farms (*n* = 64), medium farms (*n* = 64) and small farms (*n* = 64), 85 samples were collected from broiler wet meat markets (*n* = 17 from each district), and 18 samples were collected from three different supermarkets in Dhaka city (*n* = 6 from each supermarket). Note here that each breast muscle, bone marrow and composite sample was the composite mixture of four bird samples. The samples were taken from freshly slaughtered broilers in farms and broiler wet meat markets, while chilled samples were taken directly from supermarkets. About 500 g of each sample was collected into a separated sterile plastic zipper bag with proper labeling and transferred in an ice box to the Quality Control Laboratory, Department of Livestock Services (DLS), Savar, Dhaka, Bangladesh, and in dry ice to the SGS laboratory (Standard Global Services), Chennai, India, and stored at −20 °C until processing. Washing with water was avoided to prevent possible contamination with heavy metals. The samples were collected and analyzed between January and June 2022. 

### 3.2. Ethical Approval

The study protocol was authorized by the Animal Welfare and Experimentation Ethics Committee of Bangladesh Agricultural University, Mymensingh [Approval number: AWEEC/BAU/2021(54) dated 23 December 2021].

### 3.3. Questionnaire Survey

A cross-sectional survey was conducted among broiler meat consumers in the study areas of Bangladesh (Appendix A) to assess their attitude towards the consumption of broiler meat. Structured close and open-ended questions with choices were administered to the respondents. The pre-tested questionnaire was translated in Bengali (please see the Appendix A) and tested by previous researchers [2,20,21] Ninety-five questionnaires (95) were administered to broiler meat consumers across the five districts that represent five divisions of Bangladesh (19 participants from each study area). The study sites were chosen because these areas have a high number of broiler farms based on information from the Department of Livestock Services. The farms were selected based on a list of broiler farms from the districts compiled by the district livestock offices. The broiler meat consumers were selected randomly, and their consent was sought to be included in this study. Once they accepted to participate in the study voluntarily, we collected data from them via face-to-face interviews. The convenience sampling method was used to collect the data for the study where farms and samples were available [22].

### 3.4. Sample Testing

Based on our previous survey data, 10 commonly used antibiotics [2] and 3 heavy metals were quantified in the collected samples. The samples were sent to the SGS Laboratory (Standard Global Services), Chennai, India, for analyzing seven antibiotics (enrofloxacin, ciprofloxacin, neomycin, tylosin, colistin, amoxicillin and sulfadiazine) by using liquid chromatography–mass spectrometry (LC-MS/MS) (Agilent 6460 with 1290 Infinity HPLC., Woodridge, IL, USA). In addition, the samples were also sent to the Quality Control Laboratory, Department of Livestock Services, Savar, Dhaka, Bangladesh, for analyzing the remaining three antibiotics (chloramphenicol, oxytetracycline, doxycycline) by using LC-MS/MS (Nexera LCMS-8060, Shimadzu Corporation, Kyoto, Japan) and three heavy metals (arsenic, chromium and lead) by using atomic absorption spectrophotometry (AAS) (AA-7000, Shimadzu, Kyoto, Japan).

### 3.5. Sample Preparation

Weighed portions (chicken meat: 1 ± 0.01 g) of blended/minced tissues were placed into 50 mL screw-capped plastic falcon tubes separately. Then, 10 mL of diluent was added to each tube, and tubes the were shaken and vortexed for 20 min. QuechERS salt (2.0 g MgSO_4_ + 0.5 g NaCl) was added, and the mixture was vortexed for 10 min and centrifuged at 7000 rpm for 10 min at 25 °C temp. We transferred 5 mL of supernatant (the upper layer) into a 15 mL falcon tube and then added 7 mL of n-hexane, shaking and vortexing for 5 min. Two clear layers were created, and the upper layer was discarded. Then, 1 mL was taken from the lower layer and placed into a Q-sep QueChERS dSPE tube (150 mg MgSO_4_ + 50 mg PSA + 50 mg C I 8-EC), shaken and vortexed for 5 min and centrifuged at 7000 rpm for 15 min at 25 °C temp. The upper portion was taken, filtered with a 0.22 µm PVDF filter and transferred into an LC vial. 

### 3.6. Sample Extraction Procedure

Weighed portions (poultry meat: 2 ± 0.01 g) of blended meat samples were placed in 50 mL screw-capped plastic falcon tubes separately. Negative controls and recovered samples were included and presented as negative control samples for QA. Spike standards were added, and 100 µL of a 20 µg/kgD5-CAP working internal standard solution (7.4.4.4) was added to all tubes; the mixture was vortexed for 5 min and left to stand for 15 min. Then 10 mL ethyl acetate was added, and the solution was vortexed for 10 min and centrifuged at 6500 rpm for 10 min at 10 °C temp. The upper layer (ethyl acetate-5 mL) was collected, and the same procedure was repeated a second time. The upper layer (ethyl acetate-5 mL) was collected and combined with the first one in the same tube. The solvent (ethyl acetate) was evaporated under N_2_ gas at 40 °C, and the sample was reconstituted with 2 mL of 50% ACN (ACN: H_2_O: 50:50). Vortexing was conducted for 3 min followed by centrifugation for 5 min at 10 °C. The supernatant was collected, transferred to another tube and filtered with a 0.22 µm PVDF filter. The sample was transferred to a vial for analysis by LC-MS/MS. 

### 3.7. Data Analysis

All the results were entered into the Statistical Package for the Social Sciences (SPSS) software version 26.0 and analyzed accordingly. The data were then inserted in an MS Excel spreadsheet (Microsoft Excel 2018, Microsoft Corp, Redmond, WA, USA) for cleaning, processing and analysis. The data indicated about the amount of antibiotics and heavy metals present in broiler meat, expressed in parts per million (PPM). The data were examined using IBM SPSS Statistics (IBM Corp. Released 2017. IBM SPSS Statistics for Windows, Version 26.0. Armonk, NY, USA: IBM Corp) [23].

### 3.8. Pearson Correlation

The Pearson correlation method is the most common method used for numerical variables; it assigns a value between −1 and 1, where 0 indicates no correlation, 1 a total positive correlation, and −1 a total negative correlation. In addition, a correlation value of 0.7 between two variables would indicate that a significant and positive relationship exists between the two. A positive correlation signifies that if variable A increases in value, then B will also increase, whereas if the correlation is negative, if A increases, B decreases [24]. The correlation between the levels of antibiotics and heavy metals present in broiler meat overall in Bangladesh and in different locations in Bangladesh was analyzed by using IBM SPSS Statistics (IBM Corp. Released 2017. IBM SPSS Statistics for Windows, Version 26.0. Armonk, NY, USA: IBM Corp) [23]. The correlation results help an analyst to understand the relationships and their strength between variables, such as, in this study, the levels of antibiotics and heavy metals in broiler meat.

## 4. Discussion

Today, broiler farming is a booming industry all over the world. Maintaining its quality is fundamental to safely feed consumers worldwide. Farmers, poultry integrators, feed miller businesses, etc., frequently add antibiotics, enzymes, feed additives, medicines, etc., to poultry diets as part of these above-mentioned activities, which is most likely to increase the load of heavy metals in biological systems. Furthermore, a study from Bangladesh revealed the presence of antibiotic residues and heavy metals at public health-concerning levels in poultry feed, meat and eggs [25,26,27,28]. To minimize the harmful heavy metals load in the food chain, which could result in preserving meat quality, food safety and consumer health globally, holistic approaches based on various strategies are crucial.

The food chain and poultry origin food products are most likely to be contaminated by the effluent discharges caused by rapid urbanization, fast industrialization, the use of inorganic fertilizers in crop fields, unplanned lead acid rechargeable batteries recycling, emissions of transport and dumping waste from municipal and industrial areas, contamination of water bodies, etc. [29], which can be detrimental to both the environment and public health. Industrial wastes discharged from textile and tannery factories contain hazardous heavy metals above the normal levels. Both humans and the ecosystem may be harmed by the overabundance of these toxic heavy metals in the food chain. In addition, these contaminants bioaccumulate in the human and animal body, leading to serious consequences [30,31]. Heavy metals such as arsenic, chromium and lead in broiler meat might exhibit toxicity and a carcinogenic effect even at low concentration following a regular consumption of contaminated poultry products [29]. These heavy metals are also recognized as human carcinogenic agents by the International Agency for Research on Cancer and the United States Environmental Protection Agency [5]. According to earlier research, heavy metals can easily enter the food chain through water, soil or even feed, and their intake above the permissible limit may be linked to deleterious health impacts [32].

It is clear that all the analytical values of As and Cr, but not of Pd, in broiler edible tissues from different farms, broiler wet meat markets and supermarkets in Bangladesh found in this study were lower than the maximum permissible limits (MPL) of contaminants in poultry meat, as per the suggestions given by the WHO, FAO and European Commission. The high Pb levels found in the edible tissues and bone of broilers may be the consequence of feed and water contamination. The findings indicate that Pb played a significant role as an environmental pollutant in Bangladesh during the observation period. Previous studies reported that the MPL for AS, Cr and Pd are 40 µg/kg, 1000 µg/kg and 100 µg/kg, respectively, in poultry meat, and these values are considered harmless [Source: IAEA, FAO, EU standards] [33]. The values beyond these MPL are liable to cause toxicity or public hazard. The current investigation showed that, in terms of toxicity level, our analytical values of the heavy metal concentrations in broilers edible tissues indicate that poultry consumption is safe. Therefore, the broiler meat analyzed can be used by consumers all over the world in a safe and trustworthy manner. 

In previous published reports, several antibiotic residues were detected in chicken meat [34,35]. The use of antibiotics as growth promoters to boost productivity by improving disease tolerance in the broiler industry is to blame for the contamination of meat with antibiotics [36]. In this regard, the analytical data from the present studies indicated the presence of oxytetracycline, doxycycline, chloramphenicol and sulphadiazine in broiler editable tissues at a level below the MRL set by the IAEA, FAO and EU standards. Even though all the antibiotic residual concentrations in broiler edible tissues were below the MRL, the results of this study are a strong indicator of the widespread antibiotic overuse in farms and live bird markets. They also suggest that poultry farmers are ignorant of or disrespectful of the laws governing the duration of the antibiotic withdrawal period in food animals and birds. Importantly, published scientific research has established that unnecessary antibiotic exposure, even at low doses over an extended period of time, has adverse effects on consumer health and can result in the development of antibiotic resistance [37]. Antibiotic-resistant bacteria in food animals and birds can have devastating effects on the human health once they enter the food production [38]. Therefore, the widespread use of antibiotics in the poultry industry has led to the presence of residuals in poultry products, which could pose health risks for consumers, including bone marrow toxicity, allergies, mutagenicity and carcinogenicity [39,40], as well as the development of resistant strains of pathogenic bacteria [41]. 

As per the guidance provided by the EC or the WHO and FAO, the assessed heavy metal concentrations found in this study were below the limit reference values. However, the mere existence of residues is more than enough to raise concerns about public health. However, the analytical results of the heavy metal content in broiler meat found in this study indicate that this meat is safe or not harmful for the general public. As a result, consumers can use broiler meat with confidence and safety. However, a further detailed assessment with routine monitoring may be conducted to ensure the quality of the meat and the safety of the broiler food origin, as not all of the Bangladeshi farms were considered to detect major heavy metal elements found in the environment as well as its sources of origin.

## 5. Conclusions

Based on the results of this study, broiler meat, bones and composites mainly contain two antibiotics (oxytetracycline and doxycycline) and three heavy metals (arsenic, chromium and lead), though below the maximum residual limits. Broiler meat from supermarkets had lower levels of antibiotics and heavy metals compared to broiler meat obtained from various types of farms and broiler wet meat markets. The presence of antibiotics residues below the MRL levels in broiler meat, bones and composites is not harmful to public health but will facilitate the development of antibiotic-resistant bacteria, which pose a serious threat to public health. Meanwhile, although arsenic and chromium, but not lead, were present in broiler meat below the MRL values, they may raise several public health concerns. Therefore, future research on the sources of antibiotics and heavy metals contamination in broiler should be intensified to help achieve a pollution-free environment and a safe broiler meat production. In order to build a healthy and safe nation, campaigns and public awareness programs regarding misinformation that spreads misconceptions about broiler meat consumption among consumers are warranted.

## 6. Limitations

This study was limited to some selected sampling areas and farms; in addition, the samples collected from supermarkets were obtained only in Dhaka city. The sample size was also small.

## Figures and Tables

**Figure 1 antibiotics-12-00662-f001:**
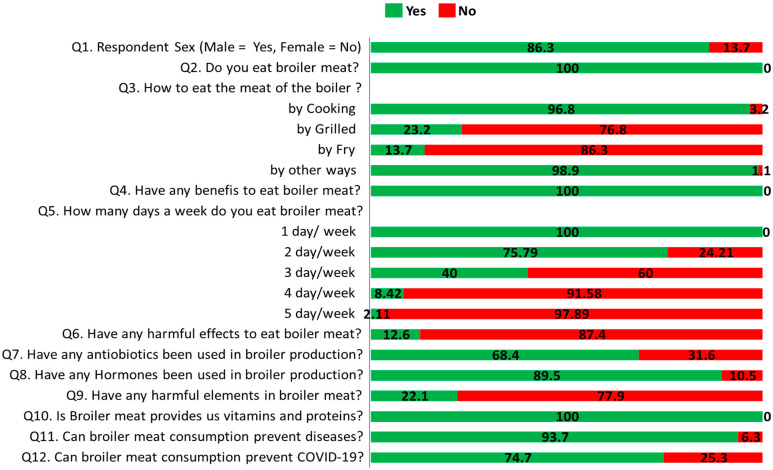
Chart showing the percentages of the responses regarding the attitude towards broiler consumption of consumers. The responses were recorded as Yes or No. Q1, question 1; Q2, question 2; Q3, question 3; Q4, question 4; Q5, question 5; Q6, question 6; Q7, question 7; Q8, question 8; Q9, question 9; Q10, question 10; Q11, question 11; Q12, question 12.

**Figure 2 antibiotics-12-00662-f002:**
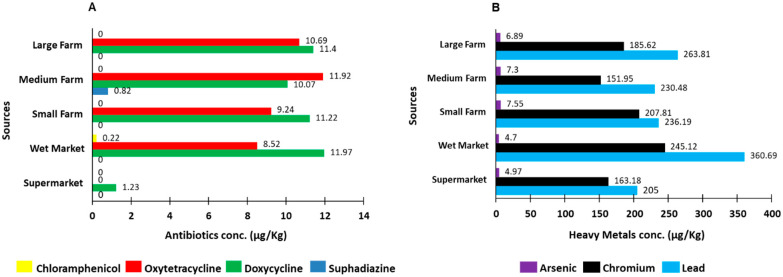
Concentration of antibiotic residues (**A**) and heavy metals (**B**) in broiler meat collected from various large farms, medium farms, small farms, broiler wet meat markets and supermarkets in Bangladesh.

**Figure 3 antibiotics-12-00662-f003:**
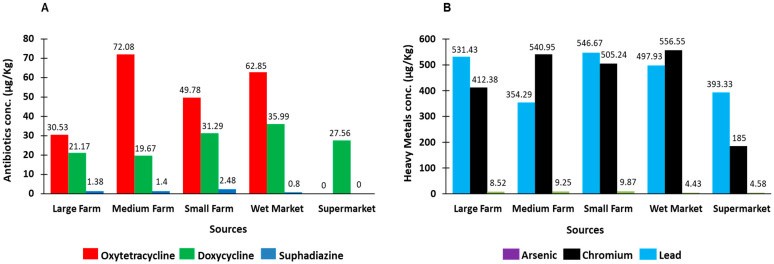
Concentration of antibiotic residues (**A**) and heavy metals (**B**) in broiler bone collected from various large farms, medium farms, small farms, broiler wet meat markets and supermarkets in Bangladesh.

**Figure 4 antibiotics-12-00662-f004:**
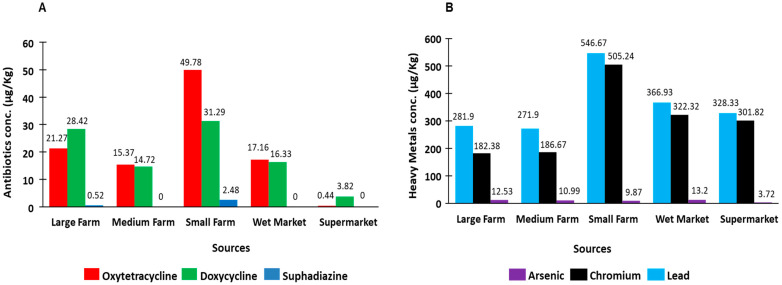
Concentration of antibiotic residues (**A**) and heavy metals (**B**) in edible broiler composites collected from various large farms, medium farms, small farms, broiler wet meat markets and supermarkets in Bangladesh.

**Figure 5 antibiotics-12-00662-f005:**
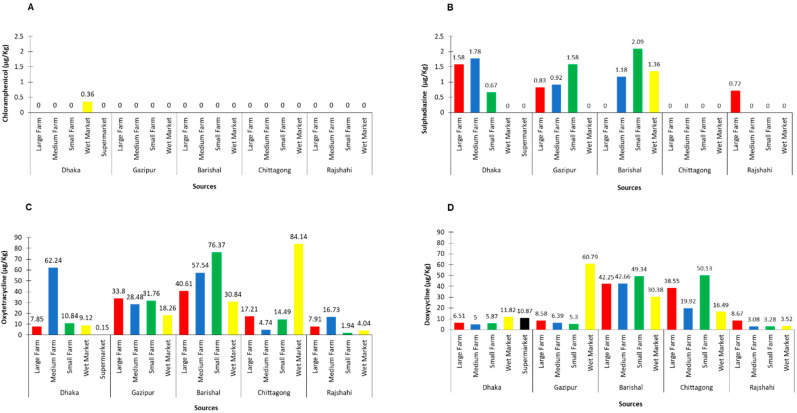
Concentration of antimicrobial residues. (**A**) Chloramphenicol, (**B**) sulphadimidine, (**C**) oxytetracycline, (**D**) doxycycline in broiler meat, bone and edible composite collected from various large farms, medium farms, small farms, broiler wet meat markets and supermarkets of the Dhaka district, Gazipur district, Barisal district, Chittagong district and Rajshahi district in Bangladesh.

**Figure 6 antibiotics-12-00662-f006:**
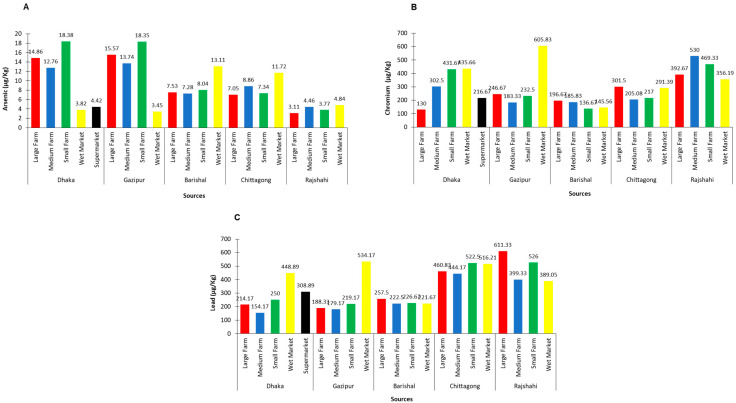
Concentration of heavy metals residue. (**A**) Arsenic, (**B**) chromium, (**C**) lead in broiler meat, bone and edible composite collected from various large farms, medium farms, small farms, broiler wet meat markets and supermarkets in Dhaka.

**Table 1 antibiotics-12-00662-t001:** Overall percentages of different antimicrobials residue in different parts of the broiler body.

Organ Samples	Positive (%)			
	Chloramphenicol	Oxytetracycline	Doxycycline	Sulphadiazine
Muscle (*n* = 98)	1 (1.02)	73 (74.49)	72 (73.47)	1 (1.02)
Bone (*n* = 98)	0 (0.00)	37 (37.76)	56 (57.14)	18 (18.37)
Composite (*n* = 99)	0 (0.00)	73 (73.74)	72 (72.73)	1 (1.01)

**Table 2 antibiotics-12-00662-t002:** Overall percentages of different heavy metals residue in different parts of the broiler body.

Organ Samples	Positive (%)		
	Arsenic	Chromium	Lead
Muscle (*n* = 98)	90 (91.84)	98 (100)	98 (100)
Bone (*n* = 98)	96 (97.96)	98 (100)	98 (100)
Composite (*n* = 99)	92 (93.88)	99 (100)	99 (100)

## Data Availability

The datasets used and analyzed during the current study are available from the corresponding author on reasonable request.

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
