# Peer review of "Assessment of Heavy Metals Contamination and Antimicrobial Drugs Residue in Broiler Edible Tissues in Bangladesh"

_antibiotics, 2023, doi:10.3390/antibiotics12040662_

Round 1

Reviewer 1 Report

Dear authors

The manuscript has assessed heavy metals contamination and antibiotic resistance in broiler meat in Bangladesh.

Moreover, the findings of this study has not demonstrated reasons for levels and types of contamination and resistance. 

I found several previous similar publications in the country which decrease the novelty of this research. 

The association or correlation of heavy metals or antibiotics does not provide interesting findings for readers as the sources or reasons are vague. 

Number of samples and studied area are narrow. 

Overall, if these queries be addressed, the manuscript can be considered for review again. 

best regards. 

Author Response

The manuscript has assessed heavy metals contamination and antibiotic resistance in broiler meat in Bangladesh. Moreover, the findings of this study has not demonstrated reasons for levels and types of contamination and resistance.

Response: we appreciate your comments, however is beyond our jurisdiction. 

I found several previous similar publications in the country which decrease the novelty of this research. 

The association or correlation of heavy metals or antibiotics does not provide interesting findings for readers as the sources or reasons are vague. 

Response: Many researchers are working on the association between antibiotic residues and heavy mental residues and their association with MDR pathogens would be a feed for those researchers and have significant important.

Number of samples and studied area are narrow. 

Response: we stated it in our manuscript as our limitation.

Overall, if these queries be addressed, the manuscript can be considered for review again. 

Response: we appreciate your valuable suggestions and have made necessary changes in our revised manuscript.

Reviewer 2 Report

Comments and Suggestions for Authors;

The article entitled as “Assessment of Heavy Metals Contamination and Antimicrobial Drugs Residue in Broiler Edible Tissues in Bangladesh” by Shaikh Mohammad Bokhtiar et al conducted an important research work to assess Heavy Metals Contamination and Antimicrobial Drugs Residue in Broiler Edible Tissues. The article is very interesting and really appreciable as the manuscript is well written, but, the followings few minor comments should be considered for the improvement of the manuscript:

1.      I think no need of figure 1.Remove it from main manuscript contents and attach supplementary file.

2.      The authors mentioned that an questionnaire was used that was tested from previous researcher. Is this questionnaire validated???Please attach the specimen of the said questionnaire at the end as supplementary file .so, the authors should clarify this questionnaire well.

3.      Please concise conclusion of yours study.

4.      Add future perspective and limitations of yours study.

5.       Cite and update the introduction with updated citations.

6.       Recheck the grammar and remove the typo mistakes.

Author Response

The article entitled as “Assessment of Heavy Metals Contamination and Antimicrobial Drugs Residue in Broiler Edible Tissues in Bangladesh” by Shaikh Mohammad Bokhtiar et al conducted an important research work to assess Heavy Metals Contamination and Antimicrobial Drugs Residue in Broiler Edible Tissues. The article is very interesting and really appreciable as the manuscript is well written, but, the followings few minor comments should be considered for the improvement of the manuscript:

  1. I think no need of figure 1. Remove it from main manuscript contents and attach supplementary file.

Response: we appreciate your valuable suggestion and transfer it to the supplementary file.

  1. The authors mentioned that an questionnaire was used that was tested from previous researcher. Is this questionnaire validated???Please attach the specimen of the said questionnaire at the end as supplementary file .so, the authors should clarify this questionnaire well.

Response: we added the questionnaires in supplementary file

  1. Please concise conclusion of yours study.

Response: we appreciate your valuable suggestions and made possible correction

  1. Add future perspective and limitations of yours study.

Response: we added future perspective and limitations of yours study.

  1. Cite and update the introduction with updated citations.

Response: we made possible correction

  1. Recheck the grammar and remove the typo mistakes.

Response: we appreciate your valuable suggestions and made correction.

Round 2

Reviewer 1 Report

Dear authors

The manuscript corrections have been made and therefore it can be accepted for publication.